# A comprehensive re-assessment of the association between vitamin D and cancer susceptibility using Mendelian randomization

Jue-Sheng Ong [1✉], Suzanne C. Dixon-Suen [2], Xikun Han [1,3], Jiyuan An[4], 23 and Me Research Team[*], Esophageal Cancer Consortium[*], Upekha Liyanage[5], Jean-Cluade Dusingize[6], Johannes Schumacher[7], Ines Gockel [8], Anne Böhmer [9], Janusz Jankowski[10,11], Claire Palles[12], Tracy O'Mara [13], Amanda Spurdle [13], Matthew H. Law [1], Mark M. Iles [14], Paul Pharoah [15], Andrew Berchuck[16], Wei Zheng [17], Aaron P. Thrift [18], Catherine Olsen [3,6], Rachel E. Neale[19], Puya Gharahkhani [1], Penelope M. Webb [20] & Stuart MacGregor [1]

Previous Mendelian randomization (MR) studies on 25-hydroxyvitamin D (25(OH)D) and cancer have typically adopted a handful of variants and found no relationship between 25(OH)D and cancer; however, issues of horizontal pleiotropy cannot be reliably addressed. Using a larger set of variants associated with 25(OH)D (74 SNPs, up from 6 previously), we perform a unified MR analysis to re-evaluate the relationship between 25(OH)D and ten cancers. Our findings are broadly consistent with previous MR studies indicating no relationship, apart from ovarian cancers (OR 0.89; 95% C.I: 0.82 to 0.96 per 1 SD change in 25(OH)D concentration) and basal cell carcinoma (OR 1.16; 95% C.I.: 1.04 to 1.28). However, after adjustment for pigmentation related variables in a multivariable MR framework, the BCC findings were attenuated. Here we report that lower 25(OH)D is unlikely to be a causal risk factor for most cancers, with our study providing more precise confidence intervals than previously possible.

[1] Statistical Genetics Group, Department of Genetics and Computational Biology, QIMR Berghofer Medical Research Institute, 300 Herston Road, Brisbane, QLD 4006, Australia. [2] Cancer Epidemiology Division, Cancer Council Victoria, 615 St Kilda Rd, Melbourne, VIC 3004, Australia. [3] Faculty of Medicine, University of Queensland, Brisbane, Australia. [4] Institute for Future Environments, Queensland University of Technology, QLD, Brisbane, QLD 4001, Australia. [5] Cancer and Population Studies Group, Population Health Department, QIMR Berghofer Medical Research Institute, 300 Herston Road, Herston, QLD 4006, Australia. [6] Cancer Control Group, Population Health Department, QIMR Berghofer Medical Research Institute, 300 Herston Road, Herston, QLD, Australia. [7] Institute of Human Genetics, Philipps University of Marburg, Marburg, Germany. [8] Department of Visceral, Transplant, Thoracic and Vascular Surgery, University Hospital Leipzig, Leipzig, Germany. [9] Institute of Human Genetics, University of Bonn, School of Medicine & University Hospital Bonn, Bonn, Germany. [10] Centre for Medicine and Health Sciences, University of Arab Emirates University, Abu Dhabi, UAE. [11] Comprehensive Clinical Trials Unit, University College London, London, UK. [12] Institute of Cancer and Genomic Sciences, University of Birmingham, Edgbaston, UK. [13] Molecular Cancer Epidemiology Group, QIMR Berghofer Medical Research Institute, 300 Herston Road, Herston, QLD, Australia. [14] Leeds Institute for Data Analytics, University of Leeds, Leeds, UK. [15] Department of Public Health and Primary Care, University of Cambridge, Cambridge, UK. [16] Department of Obstetrics and Gynecology, Division of Gynecologic Oncology, Duke University Medical Center, Box 3079 Durham, NC 27710, USA. [17] Division of Epidemiology, Vanderbilt Epidemiology Center and Vanderbilt-Ingram Cancer Center, Vanderbilt University School of Medicine, Nashville, TN 37232, USA. [18] Section of Epidemiology and Population Sciences, Department of Medicine, and Dan L Duncan Comprehensive Cancer Center, Baylor College of Medicine, Houston, TX, USA. [19] Cancer Aetiology and Prevention Group, Population Health Department, QIMR Berghofer Medical Research Institute, 300 Herston Road, Herston, QLD, Australia. [20] Gynaecological Cancer Group, Population Health Department, QIMR Berghofer Medical Research Institute, 300 Herston Road, Herston, QLD, Australia. *Lists of authors and their affiliations appear at the end of the paper. ✉email: juesheng.ong@qimrberghofer.edu.au

Vitamin D is an essential fat-soluble vitamin primarily created when the skin is exposed to ultraviolet radiation. In addition to maintaining musculoskeletal health, animal models and in-vitro studies suggest vitamin D has anti-cancer properties, as vitamin D receptors can regulate growth and apoptosis of tumour cells[1,2]. Vitamin D, whether produced in the skin or consumed, undergoes 2 hydroxylation steps to produce the active form. The first of these produces 25-hydroxyvitamin D (25(OH)D) which can be measured to determine vitamin D status. Given that serum 25(OH)D levels in the body can be readily modified through supplementation, there has been great interest in evaluating the role of vitamin D in cancer prevention[3,4].

The relationship between vitamin D and cancer continues to be debated[5–9]. Controlling for confounding in observational studies is challenging, so several studies have used Mendelian randomization (MR) to investigate whether genetic evidence support a causal role for 25(OH)D levels on cancers. Since genetic variants are randomised at meiosis, MR-based inference of the relationship between 25(OH)D and cancers via genetic proxies is less likely to be affected by reverse causality and confounding. Initial MR studies have offered some insights into a potentially causal relationship, although all used only a handful of SNPs and since some different studies have reached contradictory conclusions[10,11], there is a need for greater clarity on the issue. For most cancers, findings from MR support a null relationship between 25(OH)D and cancer risk[12], but the confidence intervals remain wide for many cancers and it is not possible to exclude clinically relevant causal effects. All previous studies relied on at most 6 SNPs associated with 25(OH)D; these cumulatively explain ~2% of variation in 25(OH)D[13–17]. While the biological pathways linking these variants to serum 25(OH)D are generally well understood, if more instruments were available, these would explain a larger portion of the phenotypic variance in 25(OH)D, while recently developed multi-instrument based MR sensitivity analyses (such as MR-Egger or median-based approaches) would provide additional assurance that the MR assumptions are not violated[18]. In particular, it is difficult to assess the potential for bias due to residual pleiotropy with a small number (i.e. <10) of variants. Moreover, the magnitude and scale of the SNP-25(OH)D associations differ among MR studies for 25(OH)D and cancer depending on the population studied in the genome-wide association study (GWAS) datasets used to probe these genetic association (i.e. differences in the relationship between genetically predicted 25(OH)D and cancer risk might be induced by variation in the SNP-25(OH)D effect sizes across sub-European ancestries). These limitations complicate the interpretation of MR findings on 25(OH)D and cancer risk published to date.

The recent release of individual-level data on serum 25(OH)D concentration for more than 400,000 people in the UK Biobank (UKB) provided an avenue for these issues to be revisited (http://www.ukbiobank.ac.uk/wp-content/uploads/2013/11/ukb_biomarker_panel_website_revised_June15.pdf). The GWAS on 25(OH)D in the UK Biobank is approximately four times larger than any previous study and would be expected to dramatically increase the number of candidate 25(OH)D instruments for the multi-SNP approach (see also the very recently published 25(OH)D GWASs[19,20]), leading to a well-powered MR analysis while enabling a wider arsenal of MR techniques[21–25] to address issues of horizontal pleiotropy, assess mediating pathways, and control for genetic heterogeneity among instruments.

In this work, we present findings from the MR analysis for the association between 25(OH)D and the risk of several common cancers using >60 25(OH)D-associated SNPs. We also report findings from Multivariable MR[25] approaches that were used to

account for related exposures, e.g. skin tanning and pigmentation that may act through sun exposure on skin cancers. We finally compare our revised findings with those reported from earlier MR studies.

## Results

In the UKB 25(OH)D GWAS, we identified 74 independent genetic instruments for 25(OH)D which together explained close to 4.0% of the phenotypic variation in serum 25(OH)D. All of the 25(OH)D variants used in previous MR analyses were successfully replicated, with comparable SNP-25(OH)D effect sizes (see Supplementary Table 1). We also identified one 25(OH)D variant located near the HAL gene, which was previously reported to influence the risk of skin cancers[26]. We removed this variant from our MR analyses on skin cancers, although the variant only explained 0.05% of 25(OH)D variation. Based on the estimated proportion of 25(OH)D variance explained by SNP of 4.0%, the power to detect moderate effect sizes (OR of 1.2 or more per 1 standard deviation (SD) increase in 25(OH)D level) is adequate for most cancers (Supplementary Table 2). In the UKB, a 1SD increase in 25(OH)D levels roughly translates to a 20-nmol/L increase in serum 25(OH)D level, which is the upper bound for the amount of 25(OH)D attainable through vitamin D supplementation (i.e. an increase of ~1000 IU/day)[27]. Sample sizes for each of the cancers evaluated are shown in Table 1. The estimated OR per 1 SD increase in genetically predicted 25(OH)D on cancer risks using the traditional 25(OH)D SNP instruments ($n = 6$) and the larger set of SNP instruments identified from the UK Biobank ($n = 79$) are shown separately in Table 2. Consistent with the MR findings based on traditional genetic instruments, we found no evidence for a causal association between 25(OH)D and the risk of breast, prostate, lung, melanoma and Barret's oesophagus/oesophageal adenocarcinoma combined (BEEA). However, higher 25(OH)D concentration was associated with reduced risk of epithelial ovarian cancer (EOC) (OR 0.89 [95% CI 0.82–0.96]), with a point estimate similar to those obtained from previous studies but with a narrower confidence interval. Subtype analyses for cancers with available data did not detect strong heterogeneity of effect sizes between subtypes. Evaluation of MR associations under various other MR models (MR-Egger, MR PRESSO, MR mode- and median-based models; see Supplementary methods) showed consistency with the original inverse-variance weighted (IVW) estimates, providing evidence against bias due to horizontal and/or directional pleiotropy (Fig. 1). The comparison of our MR findings with those obtained from previous work is shown in Table 3.

It can be challenging to interpret the univariable MR association between 25(OH)D and skin cancers (i.e. melanoma and keratinocyte cancers (KC)) given the potential confounding pathways through pigmentation and skin aging which are established risk factors for these cancers[28]. The univariable MR associations suggest higher 25(OH)D levels increase risk of basal cell carcinoma (BCC) (OR = 1.16 [1.04–1.28], while the OR for melanoma was smaller with confidence intervals overlapping 1 (OR 1.05 [0.90–1.23]). Effect sizes were widely consistent when we adjusted our MR model through a multivariable MR framework accounting for the pathway linking the 25(OH)D SNPs with childhood sunburn episodes and skin colour (Table 4). The effect estimates for melanoma (OR 1.16 [0.94–1.43]) and SCC (OR 1.08 [0.89–1.31]) in the multivariable model increased very slightly, though the confidence interval overlapped the null. For SCC, the multivariable adjusted estimate for vitamin D showed minor attenuation towards the null (OR 1.15 [0.99–1.32]). We note that the strength of the adjustment is hindered by the lack of accuracy in self-reported data and we also omitted the time spent outdoors phenotypes due to lack of power. MR estimates obtained using the conventional approach of

**Table 1 Distribution of cases and controls available for the vitamin D and cancer risk MR analysis using summary statistics from various sources.**

| Cancers | Number of controls | Number of cases | Source (PMID) | Studies |
|---|---|---|---|---|
| Barrett's oesophagus (BE) and oesophageal cancer | 17,159 | 10,279 | 27527254 | BEACON + BONN + OXFORD (BE only) + CAMBRIDGE |
| Barrett's oesophagus (BE) | 17,159 | 6167 | 27527254 | |
| Oesophageal cancer | 17,159 | 4112 | 27527254 | |
| Breast cancer | 105,974 | 122,977 | 29059683 | BCAC |
| ER + breast cancer | 105,974 | 69,501 | 29059683 | |
| ER- breast cancer | 105,974 | 21,468 | 29059683 | |
| Endometrial cancer | 108,979 | 12,906 | 30093612 | ECAC |
| EC endometrial cancer | 46,126 | 8758 | 30093612 | |
| NEEC endometrial cancer | 35,447 | 1230 | 30093612 | |
| Lung cancer | 15,861 | 11,348 | 24880342 | ILCCO (MR-Base) |
| Lung adenocarcinoma | 14,894 | 3442 | 24880342 | |
| Squamous cell lung cancer | 15,038 | 3275 | 24880342 | |
| Melanoma skin cancer | 26,409 | 15,990 | 26237428 | GenoMEL |
| Keratinocyte cancers | | | | QSKIN, 23andMe |
| Squamous cell carcinoma | 285,355 | 7400 | 31174203 | |
| Basal cell carcinoma | 279,049 | 14,940 | 31174203 | |
| Neuroblastoma | 3254 | 1627 | 23222812 | MR-Base |
| Ovarian cancer | 40,941 | 25,509 | 28346442 | OCAC |
| Clear cell | 40,941 | 1366 | 28346442 | |
| Endometrioid | 40,941 | 2810 | 28346442 | |
| High-grade serous | 40,941 | 13,037 | 28346442 | |
| Low-grade serous | 40,941 | 1012 | 28346442 | |
| Mucinous | 40,941 | 1417 | 28346442 | |
| Pancreatic cancer | 1939 | 1896 | 19648918 | PanScan (MR-Base) |
| Prostate cancer | 61,106 | 79,148 | 29892016 | PRACTICAL |

ER refers to oestrogen receptor status. EC and NEEC endometrial cancer refer to the endometrioid and non-endometrioid endometrial cancer subtypes. BE refers to Barrett's oesophagus.

excluding variants showing evidence of pleiotropic association with these risk factors were similar showing no clear relationship between vitamin D and skin cancer but with wider confidence intervals (Supplementary Table 3).

We then examined whether the MR approach suggested a link from cancer risk to altered 25(OH)D (which could hypothetically happen in observational studies if a person had lower vitamin D levels due to chronic ill health as a consequence of reduced sun exposure). For most cancers, we found little evidence for an effect of higher genetic liability to cancers altering serum 25(OH)D levels, apart from for SCC although the resultant effect estimate was negligibly small (beta = 0.01, se = 0.004; SD change in 25(OH)D per doubling of odds of SCC) as shown in Supplementary Table 4. Results for individual subtypes in the reverse MR analyses are not reported, as the statistical power was too low to detect meaningful associations.

**Sensitivity analyses.** Whilst the implementation of more SNP instruments would usually translate to better trait prediction, and hence better power for MR, it also has the potential to introduce higher levels of heterogeneity amongst the effects of the genetic instruments. Using the computed cochran Q statistics for each trait-pair evaluated in the main analyses, 7 out of 19 associations reveal moderate evidence of heterogeneity among effect sizes estimated by the 25(OH)D instruments. However, the IVW point estimates for these heterogenous trait associations were not meaningfully different from those estimated via pleiotropy-robust techniques (such as MR-PRESSO). To validate the 25(OH)D and melanoma findings, the estimated OR for melanoma after removing potentially pleiotropic variants that are associated with sunburn and pigmentation-related traits was 1.02 (0.90–1.16) which is not meaningfully different from the original findings (1.09 [0.97–1.23]; Supplementary Table 3). Using sex-specific instruments did not

change the null 25(OH)D inference for breast, prostate, endometrial, and the inverse association with ovarian cancer (Supplementary Table 5). We finally evaluated our MR associations using the recently published Revez et al.[19] 25(OH)D genetic instruments that accounts for vitamin D supplementation use - our findings were essentially unchanged (Supplementary Table 6).

## Discussion

Overall, our large multi-instrument approach strengthens confidence in previously established null findings for most cancers, except for ovarian cancer[10]. With more than seventy independent 25(OH)D instruments validated in the UKB GWAS explaining ~4% of variation in 25(OH)D, our revised MR estimates have greater robustness compared to previous MR findings between 25(OH)D and individual cancers by enabling estimation of the MR association using various MR methods and sensitivity analyses, aiding the triangulation of (or the lack of) causality. Our MR finding on endometrial cancer showed little or no effect of lower 25(OH)D influeincg endometrial cancer risk. For BCC there was an initial positive association, although this was attenuated in multivariable and sensitivity analyses Fig. 1.

For breast and prostate cancer, our estimates were similar to those presented in Jiang and colleagues[29], supported with similar estimates obtained from the other MR sensitivity models (Table 3). Our power to evaluate the MR association between pancreatic, lung cancer and neuroblastoma remained relatively poor due to the limited number of cases present in the MR-base repository. Some previous studies considered 25(OH)D on the untransformed scale and some used the log scale. Supplementary Table 7 provides a comparison of our estimates on the two scales; broadly speaking the results are qualitatively similar.

For ovarian cancer, Yarmolinsky et al.[10] previously reported a non-significant result but when we add additional SNP instruments,

**Table 2 Revised estimate for the association between one SD increase in genetically predicted serum 25(OH)D and cancer risk using UK Biobank 25(OH)D instruments.**

| Cancers | Revised heterogeneity-adjusted estimate using all UKB 25(OH)D instruments | | | Revised heterogeneity-adjusted estimate using all UKB 25(OH)D instruments | | | |
|---|---|---|---|---|---|---|---|
| | SNPs | OR (95% CI) | P-value | SNPs | OR (95% CI) | P-value | Detected outliers via MR-PRESSO |
| BEEA | 6 | 1.04 (0.88-1.23) | 0.65 | 76 | 0.98 (0.85-1.14) | 0.98 | 0 |
| BE | 6 | 1.12 (0.86-1.44) | 0.4 | 76 | 1.00 (0.84-1.18) | 0.97 | 0 |
| EA | 6 | 0.90 (0.72-1.13) | 0.36 | 76 | 0.97 (0.78-1.20) | 0.76 | 0 |
| Breast cancer | 6 | 1.01 (0.96-1.06) | 0.68 | 74 | 1.03 (0.97-1.09) | 0.38 | 5 |
| ER + breast cancer | 6 | 1.00 (0.95-1.07) | 0.88 | 74 | 1.04 (0.97-1.12) | 0.3 | 5 |
| ER- breast cancer | 6 | 1.00 (0.91-1.09) | 0.94 | 74 | 1.01 (0.92-1.12) | 0.38 | 5 |
| Endometrial cancer | 5 | 0.92 (0.81-1.05) | 0.24 | 75 | 0.95 (0.83-1.09) | 0.46 | 2 |
| EC | 5 | 0.93 (0.79-1.08) | 0.33 | 75 | 0.93 (0.81-1.08) | 0.36 | 1 |
| NEEC | 5 | 0.89 (0.61-1.31) | 0.55 | 75 | 1.02 (0.76-1.36) | 0.91 | 0 |
| Lung Cancer | 5 | 1.16 (0.88-1.54) | 0.29 | 65 | 0.94 (0.78-1.13) | 0.5 | 0 |
| LAC | 5 | 1.17 (0.80-1.72) | 0.41 | 65 | 0.91 (0.67-1.18) | 0.462 | 0 |
| SCLC | 5 | 1.03 (0.83-1.29) | 0.77 | 65 | 0.97 (0.76-1.26) | 0.843 | 0 |
| Melanoma | 5 | 1.04 (0.89-1.20) | 0.64 | 69 | 1.05 (0.90-1.23) | 0.55 | 1 |
| NMSC SCC^ | 6 | 0.96 (0.80-1.15) | 0.64 | 77 | 1.02 (0.88-1.19) | 0.77 | 0 |
| NMSC BCC^ | 6 | 1.08 (0.92-1.28) | 0.35 | 77 | 1.16 (1.04-1.28) | 0.01 | 1 |
| Neuroblastoma | 2 | 0.62 (0.27-1.41) | 0.25 | 26 | 0.74 (0.42-1.29) | 0.29 | 0 |
| Epithelial ovarian cancer | 6 | 0.92 (0.83-1.02) | 0.12 | 76 | 0.89 (0.82-0.96) | 0.02 | 0 |
| Clear Cell | 6 | 0.87 (0.63-1.20) | 0.39 | 76 | 0.87 (0.64-1.18) | 0.36 | 0 |
| Endometrioid | 6 | 1.00 (0.80-1.27) | 0.97 | 76 | 0.94 (0.77-1.15) | 0.55 | 0 |
| High-Grade serous | 6 | 0.91 (0.81-1.04) | 0.16 | 76 | 0.92 (0.82-1.03) | 0.15 | 0 |
| Low-Grade serous | 6 | 1.05 (0.71-1.54) | 0.82 | 76 | 0.99 (0.71-1.37) | 0.94 | 0 |
| Mucinous | 6 | 0.94 (0.74-1.20) | 0.62 | 76 | 0.94 (0.74-1.18) | 0.59 | 0 |
| Pancreatic cancer | 2 | 1.10 (0.52-2.33) | 0.81 | 27 | 0.95 (0.54-1.69) | 0.87 | 0 |
| Prostate cancer | 5 | 1.02 (0.93-1.14) | 0.65 | 75 | 1.11 (0.93-1.33) | 0.25 | 1 |

ER refers to oestrogen receptor status. EC and NEEC endometrial cancer refer to the endometrioid and non-endometrioid endometrial cancer subtypes. Revised estimates derived using the MR-PRESSO model which corrects for heterogeneity among SNP-effect sizes and outliers. Revised OR estimates reflect a 1 SD change in genetically predicted serum 25(OH)D concentration, which roughly translates to a 20-nmol/L change in 25(OH)D in the UK Biobank. All P-values derived from z-scores are two-sided and unadjusted for multiple comparison unless otherwise stated.
BE Barrett's oesophagus. EA oesophageal adenocarcinoma. SCC squamous cell carcinoma. BCC basal cell carcinoma. LAC lung adenomacarcinoma. SCLC squamous cell lung cancer. NMSC non-melanoma skin cancer. SCC squamous cell carcinoma. BCC basal cell carcinoma.

we find a significant association. To aid understanding of the difference between the results we used the UKB-derived estimates for the six initially identified SNP instruments. For these SNPs we obtained an estimate which is quite different to that in Yarmolinsky et al.[10]. Most of the discrepancy appears to be the result of Yarmolinsky and colleagues[10] mis-coding two of the 25(OH)D-increasing alleles at rs3755967 and rs8018720 (Supplementary Table 8). Note that Yarmolinsky et al.[10] report results on the log 25 (OH)D scale, although this does not markedly influence results and most of the discrepancy is due to the allele coding issue. When we expanded the set of SNPs from 6 to 74, the point estimate remained consistent and the confidence intervals narrowed (OR changes from 0.92 [95% CI 0.83–1.02] to 0.89 [0.82–0.96]).

The study by Winsløw et al.[28] using data from the Danish cohorts of 103 084 participants reported no evidence for an association between 25(OH)D and KC, with OR of 1.11 (95% CI 0.91–1.35) for a 20-nmol/L increase in 25(OH)D. In our univariable MR findings, we found initial evidence for an association with BCC (OR 1.18 [95% CI 1.05–1.33]) but not SCC (OR 1.02 [95% CI 0.88–1.19]). When we adjust for the potential confounding pathways through traits related to sun exposure, the MR association between 25(OH)D and BCC attenuated and the confidence intervals overlapped the null. While our MR findings remain consistent with a potentially small adverse effect, there is insufficient evidence to establish vitamin D as a causal risk factor for BCC. The findings from our univariable and multivariable MR analysis on melanoma indicate no strong link between 25(OH)D and risk of melanoma, yielding similar conclusions to previous work[30].

Our present MR results on 25(OH)D and endometrial cancer was based data from the the largest endometrial cancer consortium (ECAC). Findings from a previous meta-analysis of prospective observational studies did not show strong evidence of an association between 25(OH)D and endometrial cancer risk (summary relative risk 0.85 [95% CI 0.47–1.53] for high vs low 25(OH)D levels)[31]. Two previous trials of vitamin D3 (cholecalciferol) or calcitriol supplementation found no strong association with endometrial cancer[32,33], considered in a Cochrane review[34], but were not powered to detect significant associations for cancer endpoints. The suggestive (inverse) results of the trial[32] most regularly cited by vitamin D proponents has been questioned in the literature due to several possible shortcomings in design, analysis and/or reporting[35–37]. Our MR estimate confirms findings from observational studies and randomised clinical trials (RCTs), showing little support that 25(OH)D is an important risk factor for endometrial cancer, with sufficient power to rule out moderate (i.e. OR > 1.2) effect sizes. Furthermore, our subtype analyses show limited evidence that the MR association differed between endometrioid and non-endometrioid endometrial cancer.

Our study approach has several advantages. Firstly, our discovery sample used to identify 25(OH)D instruments was very large, with a sample size close to five times larger than those obtained from previous 25(OH)D GWAS. Furthermore, the homogeneity in measurement and bio-assay resulted in a higher quality 25(OH)D phenotype as every recruitment centre in the UKB adopted the same serum extraction protocol. The UKB instruments combined explained 4.0% of the phenotypic variance, almost doubling the variance explained by 25(OH)D SNPs compared to those used in

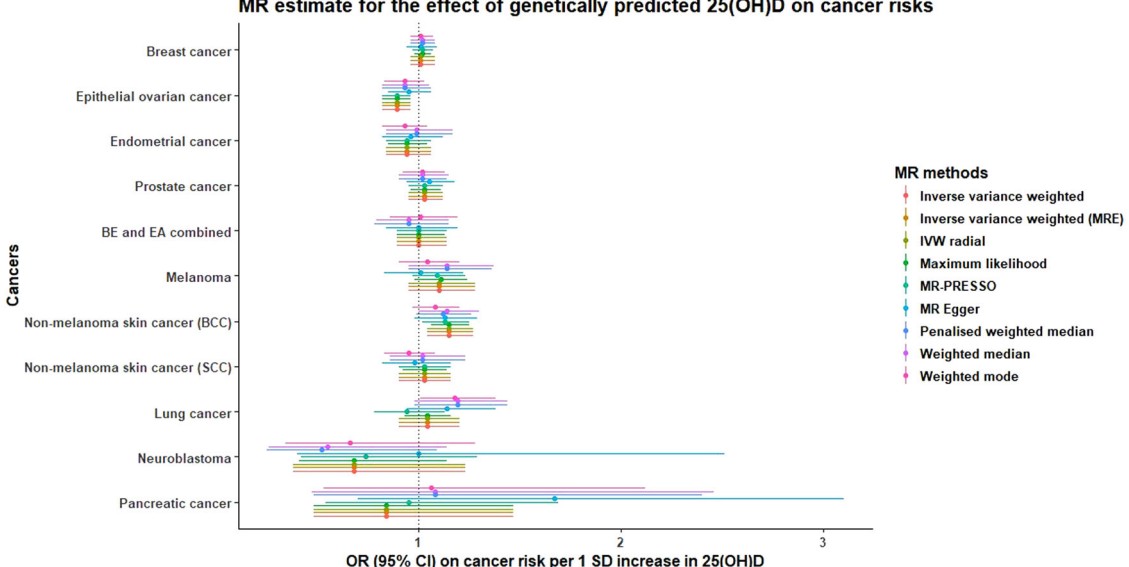

**Fig. 1 MR association between 1SD increase in genetically predicted 25(OH)D concentration and risk of cancers estimated via multiple two-sample MR model.** BE and EA combined refers to Barrett's oesophagus and oesophageal cancer combined. The *y*-axis represents the estimated OR (and the corresponding 95% CI) on individual cancers per 1 SD change in genetically predicted 25(OH)D.

**Table 3 Comparison of revised MR estimates on 25(OH)D and cancer risk against previous findings.**

| Cancers | Previous IVW MR findings | | | IVW MR estimates based on UKB 25(OH)D instruments | | | |
|---|---|---|---|---|---|---|---|
| | OR (95% CI) | Unit | *P*-value | OR (95% CI) | Unit | *P*-value | Scaling from per SD increase) |
| BEEA[63] | – | – | – | 0.98 (0.85–1.14) | 20 nmol/L | 0.82 | Multiply by 1 |
| BE | 1.21 (0.7–1.92) | 20 nmol/L | 0.41 | 1.00 (0.84–1.18) | 20 nmol/L | 0.97 | Multiply by 1 |
| EA | 0.68 (0.39–1.92) | 20 nmol/L | 0.18 | 0.97 (0.78–1.20) | 20 nmol/L | 0.76 | Multiply by 1 |
| Breast cancer[29] | 1.02 (0.97–1.08) | 25 nmol/L | 0.47 | 1.03 (0.93–1.13) | 25 nmol/L | 0.60 | Multiply by 1.25 |
| ER + breast cancer | 1.00 (0.94–1.07) | 25 nmol/L | 0.99 | 1.04 (0.93–1.16) | 25 nmol/L | 0.51 | Multiply by 1.25 |
| ER- breast cancer | 1.02 (0.90–1.16) | 25 nmol/L | 0.75 | 1.00 (0.88–1.15) | 25 nmol/L | 0.94 | Multiply by 1.25 |
| Endometrial cancer | – | – | – | 0.93 (0.80–1.07) | 20 nmol/L | 0.32 | Multiply by 1 |
| EC | – | – | – | 0.92 (0.79–1.08) | 20 nmol/L | 0.32 | Multiply by 1 |
| NEEC | – | – | – | 1.01 (0.73–1.41) | 20 nmol/L | 0.94 | Multiply by 1 |
| Lung Cancer[53] | 0.98 (0.70–1.38) | 25 nmol/L | 0.93 | 0.94 (0.78–1.13) | 25 nmol/L | 0.50 | Multiply by 1.25 |
| LAC | 0.93 (0.55–1.58) | 25 nmol/L | 0.81 | 0.91 (0.67–1.18) | 25 nmol/L | 0.46 | Multiply by 1.25 |
| SCLC | 1.02 (0.60–1.73) | 25 nmol/L | 0.95 | 0.97 (0.76–1.26) | 25 nmol/L | 0.84 | Multiply by 1.25 |
| Melanoma[30] | 1.06 (0.95–1.19) | 20 nmol/L | 0.3 | 1.09 (0.92–1.28) | 20 nmol/L | 0.31 | Multiply by 1 |
| NMSC[28] | 1.11 (0.91–1.35) | 20 nmol/L | 0.38 | – | – | – | – |
| SCC | – | – | – | 1.02 (0.88–1.19) | 20 nmol/L | 0.77 | Multiply by 1 |
| BCC | – | – | – | 1.18 (1.05–1.33) | 20 nmol/L | 0.01 | Multiply by 1 |
| Neuroblastoma[53] | 0.77 (0.22–2.70) | 25 nmol/L | 0.69 | 0.74 (0.42–1.29) | 25 nmol/L | 0.29 | Multiply by 1.25 |
| Epithelial ovarian cancer[10] | 1.02 (0.72–1.44) | Per unit log (25OHD) | 0.93 | 0.78 (0.63–0.96) | Per unit log (25OHD) | 0.03 | Use log(OH)D betas |
| Clear cell | 1.75 (0.77–3.99) | Per unit log (25OHD) | 0.18 | 0.67 (0.34–1.34) | Per unit log (25OHD) | 0.36 | Use log(OH)D betas |
| Endometrioid | 0.83 (0.46–1.50) | Per unit log (25OHD) | 0.54 | 0.87 (0.54–1.41) | Per unit log (25OHD) | 0.56 | Use log(OH)D betas |
| High-grade serous | 0.98 (0.61–1.56) | Per unit log (25OHD) | 0.92 | 0.82 (0.63–1.07) | Per unit log (25OHD) | 0.15 | Use log(OH)D betas |
| Low-grade serous | 0.64 (0.24–1.75) | Per unit log (25OHD) | 0.39 | 0.95 (0.44–2.08) | Per unit log (25OHD) | 0.99 | Use log(OH)D betas |
| Mucinous | 1.27 (0.56–2.87) | Per unit log (25OHD) | 0.57 | 0.86 (0.50–1.50) | Per unit log (25OHD) | 0.58 | Use log(OH)D betas |
| Pancreatic cancer[53] | 1.31 (0.75–2.33) | 25 nmol/L | 0.35 | 0.93 (0.46–1.92) | 25 nmol/L | 0.99 | Multiply by 1.25 |
| Prostate cancer[29] | 1.00 (0.93–1.07) | 25 nmol/L | 0.99 | 1.07 (0.89–1.29) | 25 nmol/L | 0.46 | Multiply by 1.25 |

ER refers to oestrogen receptor status. EC and NEEC endometrial cancer refer to the endometrioid and non-endometrioid endometrial cancer subtypes. The linear conversion factor is a scaling parameter to align estimates to the same scale used in previously reported estimates. Here we assume a one SD change in 25(OH)D is approximately equivalent to a 20-nmol/L change based on the distribution of 25(OH)D in the UK Biobank cohort. For ovarian cancer histotypes, we used the beta estimate obtained from the log(25(OH)D GWAS (instead of rank-transformed 25(OH)D) to draw a direct comparison with previous findings reported for per unit increase in log(25(OH)D). All *P*-values derived from *z*-scores are two-sided and unadjusted for multiple comparison unless otherwise stated. *BE* Barrett's oesophagus, *EA* oesophageal adenocarcinoma, *SCC* squamous cell carcinoma, *BCC* basal cell carcinoma, *LAC* lung adenomacarcinoma, *SCLC* squamous cell lung cancer, *NMSC* non-melanoma skin cancer.

previous studies (2.4%). These 25(OH)D instruments were also found to be associated with 25(OH)D in a recently published formal 25(OH)D GWAS analysis by Manousaki et al.[20] using the same UKB data. For the previously known SNPs from the SUNLIGHT GWAS[14] our SNP instruments (in UKB) showed comparable 25 (OH)D effect sizes. Our unified framework allows for direct comparison of MR findings for various cancers, as the same set of SNP-25(OH)D effect sizes were used across all cancers. Finally, unlike the

**Table 4 Multivariable MR model on skin-related cancers adjusting for episodes of childhood sunburn and pigmentation.**

| Risk factors in multivariate MR model | Outcome | Marginal BETA | SE | P-value | OR (95% CI) per unit change in risk factor |
|---|---|---|---|---|---|
| Episodes of childhood sunburn | Melanoma | 0.33 | 0.26 | 0.21 | 1.39 (0.83–2.34) |
| Skin colour | Melanoma | 0.06 | 0.25 | 0.81 | 1.06 (0.65–1.72) |
| 25(OH)D concentration (SD unit) | Melanoma | 0.15 | 0.11 | 0.18 | 1.16 (0.94–1.43) |
| Episodes of childhood sunburn | BCC | 2.16 | 0.18 | 5.62E-34 | 8.63 (6.09–12.21) |
| Skin colour | BCC | −0.44 | 0.16 | 7.09E-03 | 0.65 (0.47–0.89) |
| 25(OH)D concentration (SD unit) | BCC | 0.14 | 0.07 | 0.07 | 1.15 (0.99–1.32) |
| Episodes of childhood sunburn | SCC | 2.87 | 0.23 | 4.62E-35 | 17.68 (11.21–27.88) |
| Skin colour | SCC | −0.16 | 0.22 | 0.44 | 0.85 (0.56–1.29) |
| 25(OH)D concentration (SD unit) | SCC | 0.08 | 0.10 | 0.44 | 1.08 (0.89–1.31) |

Marginal BETA reflects the marginal magnitude of association between the genetic effect of 25(OH)D SNPs exert on the risk factor of interest and skin cancer outcomes after conditioning on the genetic effect on remaining risk factors. All P-values are derived from z-scores of the MR association (two-sided) and unadjusted for multiple comparison unless otherwise stated.
BCC basal cell carcinoma, SCC squamous cell carcinoma.

early 25(OH)D studies which used 2–6 SNP instruments, because we have used 74 SNPs we were able to make good use of MR sensitivity methods which rely on large numbers of SNPs (typically > 10, or ideally more) to check whether the MR assumptions made are reasonable (MR-Egger, median and mode based methods, MR-PRESSO).

While replication MR analyses using more 25(OH)D variants would generally strengthen genetic evidence for causality or a lack thereof, our study has several limitations. Firstly, any observed difference in the MR cancer estimates can be driven by differences in the 25(OH)D effect size estimates (e.g. SUNLIGHT vs UKB). We attempted to control any artefactual differences arising from scaling by expressing 25(OH)D in units of SD (rank-transformed), though it is not trivial whether these rank-based SNP-25(OH)D estimates were more informative in characterising the MR effect sizes compared to log-scaled 25(OH)D. For instance, the SUNLIGHT consortium[14] included participants from various European sub-ethnicities, while the UKB vitamin D GWAS in the present analyses were mainly conducted on the white British population. It remains unclear which SNP-25(OH)D estimates were more robust against population differences, although we showed that our inferences are broadly unaffected by the choice of SNP-25(OH)D estimate. Even with >60 SNP instruments the power to detect modest causal effects remains limited for some cancers, as these SNPs only explain 4.0% of variation in 25(OH)D. Residual pleiotropy and bias due to invalid instruments remains a concern for large multi-instrument MR studies, although we now have the luxury of efficiently utilising several sensitivity MR techniques to ensure that our inferences remain robust against these biases. Adopting sex-specific instrument are conceptually more informative for sex-specific cancers (such as endometrial, prostate and ovarian), but these are unlikely to yield different findings as the genetic architecture for vitamin D in both sexes was very similar (rg = 0.95, se = 0.02).

In our BCC and SCC GWASs, one of the cohorts in the GWAS meta-analysis (QSkin) used "super-controls" which were screened to have had no prior history of treatments for KC or other actinic lesions[38]. The estimated OR might be inflated as the stringent screening protocols are more likely to selectively pick up individuals that are genetically much less likely to develop KCs compared to using non-KC individuals as controls in the average population (Supplementary Fig. 8). It remains unclear whether this selection bias inflates the GWAS effect sizes on KC, translating to higher ORs in MR studies. Finally, the multivariable MR adjustment might not have been very effective as some of the genetic effect on sun-exposure phenotypes relied on the accuracy of self-reported information.

While the purpose of this study was to appraise and re-evaluate previous findings with more data, there are some important points

to note when comparing MR-derived findings with those from intervention studies. Linearity of effect sizes remains a strong assumption for most MR models[39], where the relationship between 25(OH)D and cancer risk is assumed to be linear across the entire trait distribution. It is possible to fit MR models including non-linear terms[40] to assess whether people with very low levels of 25(OH)D are at altered risk, although in such cases our power to detect non-linear associations would be low. RCTs remain the gold standard to establish evidence for causality, although MR can help examine observational hypotheses by limiting bias due to environmental confounding and cases where RCTs are less powered to detect effects on rarer cancers. Finally, estimates derived from MR reflect a life-long (genetic) predisposition on the exposure (vitamin D levels) which is different from the relatively short term temporal change in exposure induced by supplements in clinical trials[41].

Even though the multi-instrument approach based on >60 SNPs was more informative than the six SNPs used in recent studies, the heterogeneity of the estimates among the 25(OH)D variants for some cancers (i.e. melanoma, lung, prostate, pancreatic, neuroblastoma) was slightly higher and hence resulted in wider confidence intervals. To address this concern especially for skin cancers, we recomputed the MR association using instruments defined at p-value < 5e-8 and that were not associated with skin-related traits to inspect whether the increased heterogeneity on MR estimates is due to pleiotropic effects. The difference in estimates from the filtered set of instruments was negligible and unlikely to meaningfully change our findings. Moreover, estimates from MR-PRESSO, which is designed to discard outlier instruments with strong heterogeneity on the causal estimates for each cancer type investigated, were very similar to those evaluated using the original IVW model.

In conclusion, our revised MR estimates using more variants associated with 25(OH)D reinforce the existing body of genetic evidence suggesting that 25(OH)D concentration is not associated with risk of breast, prostate, melanoma, oesophageal and lung cancer, with tighter confidence intervals around the null. This is also the first vitamin D MR finding for endometrial cancer, and shows no evidence for a causal relationship between the two. Using a large set of genetic instruments, we were able to clarify that there does appear to be an association between higher genetically predicted 25(OH)D levels and reduced risk of ovarian cancer. These findings were consistent under alternative MR models robust against bias due to pleiotropy. We detected an initial association between 25(OH)D and basal cell carcinoma, although this was attenuated in our multivariable framework adjusting for pigmentation and sun exposure. These findings do not support the widespread use of vitamin D supplementation for prevention for most cancers, although a potential beneficial effect cannot be ruled out,

especially for rarer cancers. The protective association between 25(OH)D and ovarian cancer should be explored in further studies to determine if supplementation would provide practical benefits for this particular cancer.

## Methods

**Identifying genetic instruments for serum 25(OH)D using the UK Biobank cohort.** The UK Biobank cohort (UKB) comprises close to half a million participants aged between 37 and 70 years from across the United Kingdom. Individual-level data for serum 25(OH)D concentration were obtained from the recent UK Biobank (March 2019) biochemistry data release. Preparation and cleaning of the genotype data has been previously described[42]. Of the 438,870 white British (WB) individuals identified based on genetic principal component clustering analyses[43], 401,529 had serum 25(OH)D data. We performed a genome-wide association study on serum 25(OH)D concentration to identify genetic instruments for the MR analyses of cancer risk. Serum 25(OH)D concentration was first rank-transformed (inverse normal transformation) to allow SNP-25(OH)D effect sizes to be interpreted on a standardised Z-score scale. The GWAS was run using a linear mixed model framework accounting for relatedness, implemented using BOLT-LMM v2.2[44]. We adjusted for participant age, sex, top 10 ancestral principal components and the month of serum extraction to account for seasonal variation in sun exposure. The following criteria were used to identify genetic instruments for 25(OH)D: SNP-25(OH)D association p-value < 5e-8 (to avoid weak instrument bias) and minor allele frequency (MAF) > 0.01. Instruments were then clumped using a window of 10 Mb and maximal linkage disequilibrium of $r^2 = 0.001$ between instruments to ensure that SNPs were independent. The proportion of phenotypic variance in 25(OH)D explained by SNPs was calculated to assess statistical power for the MR analyses (Supplementary methods). The functional annotation for each of the genome-wide significant SNPs after LD-clumping was done via the --annotate command in PLINK v1.9b[45] to obtain the list of nearby genes for each variant (Supplementary Data 4).

**Obtaining instruments for skin exposure traits.** For the multivariable MR analysis on skin-related cancers (details of the analysis expanded below), we also estimated the joint effect of 25(OH)D instruments along with episodes of childhood sunburn and skin colour on risk of skin cancers. These phenotypes, obtained through questionnaires, were available in the UK Biobank. In brief, we obtained self-reported data on: episodes of childhood sunburn from 331,020 and skin colour from 433,288 WB UKB participants with suitable genetic data. The complete list of the sun exposure traits considered for the multivariable MR analysis together with the phenotype definitions and preparation for GWAS on these traits are described in greater detail in Supplementary methods.

**Description of individual cancer outcomes.** Genetic summary data for each cancer were obtained from several large cancer consortia where possible, and/or from additional population-based cohort and case-control studies. Data for melanoma, endometrial, ovarian and oesophageal cancers were obtained through approved data requests to use the GWAS summary statistics granted by the respective consortia (see below). Genetic summary data for cancers that were obtained from publicly available repositories are listed in Supplementary material. The breakdown of each cancer outcome evaluated is provided in Table 1. Brief descriptions of the GWASs obtained for each cancer are provided below.

*Breast cancer.* Genetic summary data for breast cancer was obtained from Michailidou and colleagues[46], and can be obtained from the Breast Cancer Association Consortium (BCAC) online repository. The breast cancer GWAS meta-analysis was based on 122,977 cases and 105,974 healthy controls obtained from a combination of case-control studies and large population-based cohorts that took part in BCAC and several other large GWAS consortia. Genetic QC procedures are described elsewhere[46].

*Epithelial ovarian cancer.* The GWAS summary data for EOC was retrieved from the Phelan and colleagues[47] study performed using data from the Ovarian Cancer Association Consortium (OCAC). In total, the study included 25,509 women diagnosed with EOC and 40,941 control women. For the most common subtype, high-grade serous ovarian cancer (HGSOC), the GWAS was derived from 13,037 cases and 40,941 controls; the distributions of cases for remaining subtypes are shown in Table 1.

*Endometrial cancer.* We used the endometrial cancer GWAS summary statistics reported in O'Mara and colleagues[48] which were derived from 12,906 cases and 108,979 controls across various European studies that are part of the international Endometrial Cancer Association Consortium (ECAC). Specific details on the genotyping and QC procedure were presented in the original GWAS article.

*Prostate cancer.* The GWAS summary statistics for prostate cancer were obtained from Schumacher and colleagues[49] based on data from the PRACTICAL consortium. The prostate cancer GWAS included 79,148 cases and 61,106 controls from participants of European ancestry who were recruited for studies from around the world.

*Melanoma and keratinocyte cancers.* The melanoma GWAS dataset was obtained from the 2015 melanoma GWAS meta-analyses[50] comprising 12,874 cases and 23,203 controls from populations of European ancestry. For keratinocyte cancers (KC), we performed separate fixed effect inverse-variance-weighted squamous cell carcinoma (SCC) and basal cell carcinoma (BCC) meta-analysis of GWAS summary data available in Liyanage and colleagues[51], derived from 23andMe, Inc., a personal genetics company, and the QSkin[38] cohort. The UKB samples were excluded to ensure non-overlapping samples in the two-sample MR framework (see below). QSkin data were collected from participants of European ancestry currently residing in Queensland, Australia and consist of 1995 BCC cases, 821 SCC cases and 4797 controls[38]. Detailed information about genotyping and quality control of QSkin BCC and SCC GWASs and cleaning of the 23andMe data were provided previously[51], and the cohort description is outlined in supplementary methods.

*Oesophageal cancer.* Barret's oesophagus (BE) is a precursor condition that is strongly associated with oesophageal adenocarcinoma (EA). Given the strong genetic correlation between the two diseases[52], we decided to combine both of these diseases into a single outcome (BEEA) to maximise the number of cases. For BEEA (including BE and EA separately), the GWAS findings were previously reported in Gharahkhani and colleagues[52]; the GWASs were conducted using 6167 BE patients, 4112 EA patients and 17,159 controls, all of whom were of European ancestry and were recruited from 15 epidemiologic studies from Australia, Europe and North America as part of the Barrett's and Oesophageal Adenocarcinoma Consortium (BEACON); and studies from Born, Germany; Oxford and Cambridge in the United Kingdom. Specific details on the study participants, genotyping and imputation were previously reported[52]. GWAS summary statistics for the risk of BEEA combined were used for the main analysis; we retained the summary statistics for BE and EA separately for sensitivity analyses.

*Datasets from MR-base for lung cancer, pancreatic cancer and neuroblastoma.* we did not have access to GWAS summary statistics generated from large consortia for other cancers including lung, pancreatic and neuroblastoma[53]. We relied on the curated dataset available from the MR-Base online platform[54] to perform a two-sample MR analyses using the UKB 25(OH)D instruments. The corresponding study reference for each of the cancer datasets and their respective sample sizes are listed in Table 1. We applied a similar approach to evaluate the MR association under multiple MR estimators as previously discussed.

**Two-sample MR analysis.** The two-sample MR framework is a powerful statistical approach that allows for causal inference analyses in the absence of individual-level data, as it only requires genetic estimates obtained via GWAS summary statistics. To ensure that these inferences are not biased by reverse causality, the samples used to derive SNP-25(OH)D and SNP-cancer associations have to be independent by design[55]. LD-proxies for 25(OH)D SNPs were applied (LD $r^2$ with tagged 25(OH)D SNP > 0.8) where the SNPs of interest were not well-imputed/genotyped from cancer GWAS summary statistics. Prior to any MR analyses, we first performed a standard allele harmonisation to align alleles on the forward strand. Here palindromic SNPs with a non-inferrable allele frequency (MAF > 0.3) were excluded from the analysis. For each cancer trait evaluated, we first estimated the MR association between 25(OH)D and cancer using the inverse-variance weighted (IVW) estimator which combines the SNP-25(OH)D and SNP-cancer association across multiple 25(OH)D instruments. The weighted median, weighted mode and MR-Egger estimators were then applied to help triangulate causal inference in the presence of a small proportion of invalid instruments or directional pleiotropy bias[56–58]. The MR-PRESSO[22] technique was also applied to provide an MR estimate which is robust against the presence of heterogeneity among SNP effects. The estimated MR association between 25(OH)D and cancer risk was expressed in odds ratio (OR) of cancer per one standard deviation (SD) increase in genetically predicted 25(OH)D concentration. Multivariable MR adjusting for the effect of pigmentation and sun exposure on skin cancers

Darker pigmentation reduces the amount of vitamin D obtained through sun exposure[59]. Moreover, a recent study in Australian twins showed evidence of a shared genetic architecture between pigmentation traits and 25(OH)D concentration[60]. Interpreting genetic findings can hence become difficult as variants influencing serum 25(OH)D might act through changes in pigmentation/skin colour or sun exposure, which are themselves independent risk factors for skin cancers[61]. This can potentially violate the exclusion restriction MR assumption. To estimate a direct effect between genetic 25(OH)D and skin cancers not influenced by these risk factors, we first fitted multivariable MR models conditioning on factors related to proxies of outdoor activity, skin colour, hair colour, episodes of childhood sunburn, and skin aging. This is particularly important to remove potential bias due to horizontal pleiotropy. SNPs achieving genome-wide significance for each trait were included in the multivariable MR analysis (see Supplementary methods). We first evaluated the strength of the instrument in a multivariable MR setting using the conditional F-statistics (cond. F-stat); we dropped traits that did not fulfil criteria of strong instrument (i.e. cond. F-stat > 10) from the multivariable MR analysis, and to reduce collinearity only picked one trait for each category (from pigmentation-related, chronic sun exposure and time spent outdoors traits). In our final multivariable MR model we included skin colour

(cond. F-stat 11.4), episodes of childhood sunburn (cond. F-stat 13.1) and 25(OH) D (cond. F-stat 59.4) (Supplementary Table 9). The multivariable MR analysis was performed by jointly fitting the SNP-25(OH)D and SNP-skin trait effect sizes simultaneously in the weighted regression model on the SNP-cancer association. These analyses were conducted using the mv_multiple() function available in the TwoSampleMR package in R curated in the MR-Base platform[54].

**Reverse MR analysis**. We also evaluated whether there is genetic evidence for a reverse causal effect where disease (cancer) status consequently altered serum 25 (OH)D levels by performing a bi-directional MR analysis. Here, genome-wide loci for each cancer were selected and clumped as independent instruments for cancer exposure. The curation of cancer exposure instruments are elaborated in the supplementary methods. Effect estimates on 25(OH)D are expressed for a doubling of odds of cancer (i.e. multiplying the MR estimate by $\ln(2) \sim = 0.693$). The purpose of the reverse MR analyses is to assess evidence of reverse causality - a situation where higher genetic liability to cancers might have an effect on 25 (OH)D levels.

**Sensitivity analyses**. A series of MR sensitivity analyses were performed to ensure that our findings were robust against weak violation of MR assumptions. First, we evaluated whether the genetic architecture for 25(OH)D differed by sex by assessing the genetic correlation between 25(OH)D in both sexes via bivariate LD-score regression[62]. For cancers where the incidence greatly differs by sex, we repeated our analyses using 25(OH)D instruments and SNP-25(OH)D effect estimates calibrated in males and females separately. Cochran Q tests were used to evaluate the presence of global heterogeneity in MR estimates for each cancer. We also evaluated the impact of adjusting for vitamin D supplementation use by comparing our MR findings with those derived using the Revez et al. SNP instruments.

Complementing our multivariable MR analyses on skin cancers, we repeated our MR analyses using only a filtered set of variants that are not associated with skin-related traits to reduce the potential bias in MR findings for these cancers due to horizontal pleiotropy. This was done by removal of 25(OH)D SNPs that are associated ($p < 1e-5$) with pigmentation, skin aging or tanning in both the UK Biobank and QSkin cohort. MR scatter plots and funnel plots were also generated for each cancer trait evaluated to allow physical inspection of potential SNP outliers.

**Literature review of MR studies on 25(OH)D and cancers**. We performed a literature search in the PUBMED database for previously published MR findings on 25(OH)D and susceptibility to any of the cancers included. Full details on the search strategy and the search terms applied are provided in supplementary methods. For each of these cancers, we prioritise reporting the largest study to date as defined by the analysis with the largest number of cancer cases evaluated with the greatest amount of 25(OH)D variance explained by SNP instruments.

**Reporting summary**. Further information on research design is available in the Nature Research Reporting Summary linked to this article.

## Data availability

Summary data from each consortium is obtained from their respective public data repositories. The data from BCAC can be accessed here [http://bcac.ccge.medschl.cam.ac.uk/bcacdata/oncoarray/]. The GWAS summary statistics for the prostate cancer risk is available here [http://practical.icr.ac.uk/blog/?page_id=8164]. The ovarian cancer GWAS data can be obtained through written request to the OCAC program committee [http://ocac.ccge.medschl.cam.ac.uk/]. The ILCCO GWAS for lung cancer as well as the genetic summary data on neuroblastoma and pancreatic cancer (PanScan) were already deposited in the MR-Base database [http://app.mrbase.org/]. Genetic summary data for melanoma (GenoMEL) and oesophageal cancers (ECC) were provided through formal application for use of the data through the respective program committee for each consortium. Data for the SCC and BCC GWAS meta-analysis can be obtained via written request to Dr. Stuart MacGregor (email: stuart.macgregor@qimrberghofer.edu.au), excluding the 23andMe data. The individual-level phenotype and genotype data for UK Biobank can be accessed through formal application to the UK Biobank via [https://www.ukbiobank.ac.uk/]. The GWAS summary statistics for keratinocyte cancers from the 23andMe dataset will be made available through 23andMe to qualified researchers under an agreement with 23andMe that protects the privacy of the 23andMe participants. Please visit [https://research.23andMe.com/collaborate] for more information and to apply to access the data. The authors declare that all other data supporting the findings of this study are available within the paper and its supplementary information files.

## Code availability

The GWAS analysis for 25(OH)D was performed using BOLT-LMM v2.3.4 (available in http://data.broadinstitute.org/alkesgroup/BOLT-LMM/). Association analysis performed in open-source statistical software R v4.0.2 [https://www.r-project.org/] using the TwoSample Mendelian randomization R package [https://github.com/MRCIEU/TwoSampleMR]. Illustrations produced using the ggplot2 v3.2.1R package available from the R CRAN repository (https://cran.r-project.org/).

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

## Acknowledgements

We would like to thank the research participants and employees of 23andMe for making this work possible. The QSkin Study is conducted by a team of researchers from the QIMR Berghofer Medical Research Institute. We are grateful to the Queenslanders who have willingly given their time to take part in QSkin. We are grateful to the National Health and Medical Research Council of Australia (NHMRC) for funding (APP1063061, APP1185416, APP1123248 and APP1154543). This work was conducted using the UK Biobank Resource (application number 25331). The UK Biobank was established by the Wellcome Trust medical charity, Medical Research Council (UK), Department of Health (UK), Scottish Government, and Northwest Regional Development Agency. It also had funding from the Welsh Assembly Government, British Heart Foundation, and Diabetes UK. This work is primarily funded by the National Health and Medical Research Council of Australia (NHMRC) under grant application APP1063061, APP1185416, APP1123248 and APP1154543. This work was also conducted using the UK Biobank Resource (application number 25331).

## Author contributions

The study was conceived and designed by J.S.O., P.M.W. and S.M. Summary data from various studies were contributed by T.O.M., A.S., P.P., M.H.L., M.M.I., A.B., J.S., I.G., A.B., J.J., W.Z., C.P., E.C.C., 23andMe, A.P.T., C.O., R.E.N., P.G. and P.M.W.; J.S.O., S.C.D., U.L., J.C.D., X.H. and J.A. collected, analysed and interpreted the data. The first draft of the manuscript was written by J.S.O. and S.M. All other authors provided critical feedback on the results and contributed to writing the final version of the manuscript.

## Competing interests

All authors from the group 23andMe Research Team are employees of the company 23andMe, Inc. All remaining authors declare no conflict of interest.

## Additional information

## Esophageal Cancer Consortium

Rebecca Fitzgerald[21], Matt Buas[22], Marilie D. Gammon[23], Douglas A. Corley[24], Nicholas J. Shaheen[25], Laura J. Hardie[26], Nigel C. Bird[27], Brian J. Reid[28], Wong-Ho Chow[29], Harvey A. Risch[30], Weimin Ye[31], Geoffrey Liu[32], Yvonne Romero[33], Leslie Bernstein[34], Anna H. Wu[35], David E. Whiteman & Thomas Vaughan[36]

[21]MRC Cancer Unit, University of Cambridge, Cambridge, UK. [22]Roswell Park Comprehensive Cancer Center, Elm and Carlton Streets, Buffalo, NY, USA. [23]Department of Epidemiology, University of North Carolina, Chapel Hill, NC, USA. [24]Division of Research, and San Francisco Medical Center, Kaiser Permanente Northern California, Oakland, CA, USA. [25]Division of Gastroenterology and Hepatology, University of North Carolina School of Medicine, University of North Carolina, Chapel Hill, NC, USA. [26]Division of Epidemiology, University of Leeds, Leeds, UK. [27]Department of Oncology, Medical School, University of Sheffield, Sheffield, UK. [28]Department of Genetics, Fred Hutchinson Cancer Research Center, Seattle, WA, USA. [29]Department of Epidemiology, MD Anderson Cancer Center, Houston, TX, USA. [30]Department of Chronic Disease Epidemiology, Yale School of Public Health, New Haven, CT, USA. [31]Department of Medical Epidemiology and Biostatistics, Karolinska Institute, Stockholm, Sweden. [32]Pharmacogenomic Epidemiology, Ontario Cancer Institute, Toronto, ON, Canada. [33]Division of Gastroenterology and Hepatology, Mayo Clinic, Rochester, MN, USA. [34]Department of Population Sciences, Beckman Research Institute and City of Hope Comprehensive Cancer Center, Duarte, CA, USA. [35]Department of Preventive Medicine, University of Southern California/Norris Comprehensive Cancer Center, Los Angeles, CA, USA. [36]Department of Epidemiology, School of Public Health, University of Washington, Seattle, WA, USA.

## 23 and Me Research Team

M. Agee[37], B. Alipanahi[37], A. Auton[37], R. K. Bell[37], K. Bryc[37], S. L. Elson[37], P. Fontanillas[37], N. A. Furlotte[37], D. A. Hinds[37], K. E. Huber[37], A. Kleinman[37], N. K. Litterman[37], M. H. McIntyre[37], J. L. Mountain[37], E. S. Noblin[37], C. A. M. Northover[37], S. J. Pitts[37], J. Fah Sathirapongsasuti[37], O. V. Sazonova[37], J. F. Shelton[37], S. Shringarpure[37], C. Tian[37], J. Y. Tung[37], V. Vacic[37] & C. H. Wilson[37]

[37]23andMe, Inc, 223N Mathilda Ave, Sunnyvale, CA, USA.

