## [Peer Review File · Nature Communications]

REVIEWER COMMENTS

Reviewer #1 (Remarks to the Author): Expert in Mendelian Randomization

This paper uses MR to examine the effect of Vitamin D on a range of cancer types and then applies multivariable MR to test whether sun exposure may be a pleiotropic pathway that explains any observed association. Although this is an interesting and relevant question there are issues with the analysis that need to be addressed.

1. More details should be provided on the robust methods used for the univariable MR. The methods used are only robust to pleiotropy under certain assumptions and not to all potential forms of pleiotropy, this should be made clearer in the discussion of the results. These methods also aren't robust to weak instruments, however if all the variants are genome-wide significant for Vitamin D then weak instruments will not be a particular issue in the univariable MR analyses.
2. Although the multivariable MR including measures of sun exposure is a useful analysis that potentially adjusts for pleiotropic effects of the SNPs on the cancer outcomes via sun exposure, it is not clear what benefit is gained in the multivariable MR by including so many measures of sun exposure as separate exposures in the multivariable MR. The authors should consider reducing the number of measures or explain why using so many benefits the analysis.
3. The multivariable MR analyses include a large number of additional exposures but only uses genetic variants associated with Vitamin D. It is conventional in multivariable MR to include SNPs associated with each of the exposures and failure to do this increases the risk of weak instruments causing bias in the multivariable MR estimation results.[1] The authors should include additional SNPs associated with the other exposures included in their analysis and should report the conditional F-statistics to test the strength of the instruments on this setting.

More minor points

4. In the abstract the sentence "For basal cell carcinoma, the positive association from standard MR analysis was eventually attenuated after correcting for pigmentation-related variables" appears to contradict the sentence before.
5. On line 170; the multivariable MR approaches used here can only evaluate the presence of pleiotropy that acts through sun exposure. This should be made clear for readers unfamiliar with the method.

References

1. Sanderson, E., Spiller, W., & Bowden, J. (2020). Testing and Correcting for Weak and Pleiotropic Instruments in Two-Sample Multivariable Mendelian Randomisation. *BioRxiv*.

Reviewer #2 (Remarks to the Author): Expert in Mendelian Randomization and vitamin D associations

The study by Ong et al is an MR revisiting the causal role of vitamin D in 10 cancers, using an increased set of 25(OH)D SNPs as MR instruments, following a GWAS on the recently released UKBB 25(OH)D data. These SNPs explain a larger portion of the variance in 25(OH)D compared to the 6 variants used as instruments in previous MR studies. The results are largely compatible with previous MR studies, showing evidence for a causal role of vitamin D only in ovarian cancer. For some cancers, the confidence intervals of the MR estimates are tighter compared to previous studies. This is the first study reporting MR results on the association of vitamin D with endometrial cancer.

The design of the study has many strengths: the authors used different MR methods, each one with its own assumptions, to control for pleiotropy. They used multivariable MR to test the effect of traditional risk factors such as exposure to sun and skin pigmentation in certain types of skin cancers. Finally, they controlled for sex-specific effects, and reverse effects of cancers on vitamin D levels.

Major points:

My main point of critique is why the authors did you not use the already published GWAS on 25(OH)D in UKBB (Revez et al Nat Comm 2020 or Manousaki et al AJHG 2020). Both GWAS reported a much larger number of independent variants ($N \sim 140$), explaining a much larger portion of the variance in 25(OH)D levels than the 4% reported by the 74 SNPs of the present GWAS. For instance, in the Revez et al GWAS, the SNPs explain up to 13% of the variance in 25(OH)D). Also, these GWAS adjusted for vitamin D supplementation which was not included as a covariate in the GWAS performed for this study. The authors could rerun the MR analyses, and compare their original findings with the results using the SNPs and the respective betas from one of the two GWAS.

Minor points:

Abstract

Line 90: capitalize Mendelian

Line 94: unified MR analysis.

Line 99: "Unlikely to be a strong risk factor": I would replace this by "causal risk factor", since MR is a method for causal inference.

Line 130: "measured to determine vitamin D status": you could precise "whose serum levels are measured"

Line 136: "examine the issue": I would say "to investigate the causality of 25(OH)D levels in cancer"

Line 139: In line with the previous comment, I would say: "Into a potentially causal relationship"

Line 148-149: I would add: "if more instruments were available, these would explain a larger portion of the variance of 25(OH)D, while recently..."

Line 243: “alternative sensitivity MR models”: I would rephrase this as follows: “using various MR methods and sensitivity analyses”

-A statement about power analysis results in the Results section is missing (although this power analysis appears in the supplement).

Line 254, and line 317-318: The authors mention scales: what is the interpretation of a 1SD change in the exposure, and how does that interpret into nmol/l of measured 25(OH)D? Also, how does this compare to changes in serum 25(OH)D as an effect of supplementation? It would be clinically relevant to provide this information. I see a small note in a legend of Table 2 (1 SD corresponds to 20nmol/l change in 25(OH)D, but no mention in the main text.

Line 269: Please spell out KC and line 294 spell out EC (it’s the first mention of these abbreviations).

Line 311: Here is the first mention of mode-based method. You need to at least mention all 4 MR methods somewhere in the results before discussing them.

Table 3: Please add the references for the previous MR studies from which the presented estimates were extracted.

Despoina Manousaki, MD, PhD

Reviewer #3 (Remarks to the Author): Expert in GWAS and MR of vitamin D and cancer

This is a large scale MR study, aiming to test causality of association between 25(OH)D and the risk of several types of cancer. The paper represents a large effort, and conclusions appear largely justified by the data. I have some concerns related to the reporting from the new instrument for 25(OH)D.

The group has run GWAs on 25(OH)D, presumably in parallel with an other GWAS using this same data from the UKB, which was published some time ago. I appreciate that the published GWAs results (ref 35) may not have been available to the authors at the time of writing this manuscript, however, I was surprised to see running a GWAS in UKB introduced as a hypothetical possibility in the introduction rather than being included with a statement supported by a proper reference (Reference only included in the discussion).

As a key strength of the study they indicate the ability to test for bias by pleiotropy. However, to do this, they first introduce obvious pleiotropy using the new variants. There is a notable lack of transparency with respect to what this new instrument contains (only RS numbers presented, while no gene names are included) but from the formal GWAs I know it includes several well known highly pleiotropic variants such as APOE, GCKR, and CETP, which are unlikely to have a primary role related to 25(OH)D.

I do not see the use of the new instrument as a particular advance with this study, as the earlier studies would have been likely to be notably less affected by related biases due to clearly defined roles of the variants in the metabolic vitamin D pathway. Therefore, the title of the paper is misleading in making notes about an 'improved instrument' and rather than using this type of opinionated description, my suggestion would be to stick to the facts and e.g. describe the instrument by the number of SNPs.

Given uncertainties with the new instrument, I propose the authors to repeat the analyses using the old instrument. This could be informative, given the sample size for participants with cancer in their study is somewhat larger than in the earlier studies. This would also allow them to meta-analyses across any independent studies presenting information for relevant SNPs, allowing them to maximise the sample size where possible.

In power calculation tables (supplementary 2), it is not made clear what is the estimated difference in 25(OH)D that relates to the proposed ORs. Please also clarify units elsewhere, and ensure it is clear to the reader what kind of difference in 25(OH)D (optimally in units such as nmol/l) relates to each of ORs across all tables and figures.

Response Letter to Reviewers

We thank the reviewers for their comments and suggestions to help us improve our work. The complete *point-by-point* response to the reviewer's queries is below, accompanied with tables and figures in the Appendix document (attached as a separate "related-manuscript" document). Our response written with the "Times New Roman" font face and changes made to the manuscript are shown in blue.

Summary of main changes:

- 1) We have now revised our analysis to fit all the SNPs for each trait in the multivariable MR model, and reduced the number of traits in the model as suggested by Reviewer #1. The revised findings were similar to the previous model, showing weak attenuation of effect size between 25(OH)D and skin cancer once pigmentation-related variables are accounted for.
- 2) Based on Reviewer #2's suggestion, we have added an additional analysis contrasting our MR findings against those evaluated using the vitamin D variants from the recently published Revez et al. study that accounts for supplementation use - our findings remain broadly consistent.
- 3) To answer Reviewer #3's concern, we have provided further descriptions to explain the rationale for the use of more vitamin D SNPs to tackle issues of horizontal pleiotropy in MR findings, and making it clear in the main text that our analysis consists of both the MR findings based on the old instrument as well as the new and larger set of UK Biobank instruments.

RESPONSE TO REVIEWER COMMENTS

Reviewer #1 (Remarks to the Author): Expert in Mendelian Randomization

This paper uses MR to examine the effect of vitamin D on a range of cancer types and then applies multivariable MR to test whether sun exposure may be a pleiotropic pathway that explains any observed association. Although this is an interesting and relevant question there are issues with the analysis that need to be addressed.

Q1. More details should be provided on the robust methods used for the univariable MR. The methods used are only robust to pleiotropy under certain assumptions and not to all potential forms of pleiotropy, this should be made clearer in the discussion of the results. These methods also aren't robust to weak instruments, however if all the variants are genome-wide significant for Vitamin D then weak instruments will not be a particular issue in the univariable MR analyses.

R: The detailed descriptions for each univariable MR method have been shown in the supplementary methods section. We have now clarified (throughout the entire manuscript) that these sensitivity MR models are only robust towards horizontal pleiotropy (as opposed to the presumption that it is applicable for all forms of pleiotropy). We have also clarified that in our study design (using only genome-wide significant variants) weak instrument bias is unlikely to be a major concern.

For instance, the part discussing the alternative MR methods in the Result section now reads:

“Evaluation of MR associations under various other MR models (MR-Egger, MR PRESSO, MR mode- and median-based models; see Supplementary methods) showed consistency with the original inverse variance weighted (IVW) estimates, providing evidence against bias due to horizontal and/or directional pleiotropy”

And in methods:

“The following criteria were used to identify genetic instruments for 25(OH)D: SNP-25(OH)D association p-value $<5e-8$ (to avoid weak instrument bias) and minor allele frequency (MAF) >0.01 .”

Q2. Although the multivariable MR including measures of sun exposure is a useful analysis that potentially adjusts for pleiotropic effects of the SNPs on the cancer outcomes via sun exposure, it is not clear what benefit is gained in the multivariable MR by including so many measures of sun exposure as separate exposures in the multivariable MR. The authors should consider reducing the number of measures or explain why using so many benefits the analysis.

R: The issue with phenotypes on sun exposure and pigmentation in the UK Biobank is that they often rely on the accuracy of self-reported data. Self-reported measures of sun exposure can be prone to non-differential measurement error; using multiple measures partly accounts for this.

We hence anticipate that including several variables simultaneously in the model might increase the chance of capturing horizontal pleiotropy that confounds the vitamin D estimate, even though we are well aware of the problem with multicollinearity since some of these variables are strongly correlated. To use all of this information in the most robust manner would require dimension reduction techniques such as principal component analyses, which is an interesting future direction but would be outside the scope of the present study. Moreover, we agree with the reviewer that adding many traits without evaluation of instrument strength in the MVMR setting might also contribute to biased MVMR estimates due to weak instruments (when fitted altogether). As suggested by the reviewer, we have re-run the MVMR analysis with a reduced number of measures (using 1 pigmentation trait, 1 sun exposure trait and vitamin D); when this is done the resultant MVMR results were similar to those obtained previously.

Brief summary of how we selected traits in the reduced MVMR model

We first re-evaluated the MVMR association by fitting instruments for each exposure in the MVMR model (8 traits in total), and re-clumped the variants for independence. This resulted in a total of 223 SNP instruments. We then tested the strength of the instrument in the MVMR setting using the conditional F-statistics as suggested, implemented via the `strength_mvmmr()` function in the MVMR R package. Apart from vitamin D, none of the remaining seven traits had a conditional F-statistic > 10 , which might generate weak instrument bias in the resultant MVMR estimate. We hence reduced the number of traits, by prioritising traits with the highest F-statistics and one trait for each category. Because none of the “time spent outdoors” traits had good conditional F-statistics, we omitted these traits entirely from our analysis. In the reduced/revise (MVMR) model, we selected “skin colour”, “episodes of childhood sunburn” and “vitamin D”, all of which had large conditional F-statistics in the reduced SNP set ($n=209$). The comparison of the conditional F-statistic for each exposure in the original and reduced model is

shown in Appendix Table 1, and the revised estimates are shown in Appendix Table 2. We have also included similar descriptions in Supplementary methods.

We have replaced the MVMR analysis in the main manuscript with findings from the reduced model as advised; note that results from the original model can still be accessed in the supplementary section (with the caveat that these estimates might be affected by weak instrument bias). We have fully replaced the content in Table 4 in the main manuscript with MVMR estimates from the newly revised model (3 traits).

Q3. The multivariable MR analyses include a large number of additional exposures but only uses genetic variants associated with vitamin D. It is conventional in multivariable MR to include SNPs associated with each of the exposures and failure to do this increases the risk of weak instruments causing bias in the multivariable MR estimation results.[1] The authors should include additional SNPs associated with the other exposures included in their analysis and should report the conditional F-statistics to test the strength of the instruments on this setting.

R: As elaborated in our response to (Reviewer#1 Q2), we have now revised our MVMR analysis to

(i) include SNPs associated with any included exposure

(ii) reduced the number of traits in the multivariable MR analysis reported in the main text.

The conditional F-statistics for each instrument are shown in Supp Table 10 (also available as Appendix Table 1), showing evidence that the 3 traits used in our MVMR model (childhood sunburn, skin colour and 25(OH)D) satisfy the strong instrument criteria. We have also modified the Results and Methods sections to incorporate these changes.

In Methods, the relevant paragraph now reads:

“To estimate a direct effect between genetic 25(OH)D and skin cancers not influenced by these risk factors, we fitted multivariable MR models conditioning on factors related to proxies of outdoor activity, skin colour, hair colour, episodes of childhood sunburn, and skin aging. This is particularly important to remove potential bias due to horizontal pleiotropy. SNPs achieving genome-wide significance for each trait were included in the multivariable MR analysis. We first evaluated the strength of the instrument in a multivariable MR setting using the conditional F-statistics (cond. F-stat); we dropped traits that did not fulfill criteria of strong instrument (i.e. cond. F-stat > 10) from the multivariable MR analysis, and to reduce collinearity only picked one trait for each category (from pigmentation-related, chronic sun

exposure and time spent outdoors traits). In our final multivariable MR model we included skin colour (cond. F-stat 11.4), episodes of childhood sunburn (cond. F-stat 13.1) and 25(OH)D (cond. F-stat 59.4).”

Results:

“Effect sizes were widely consistent when we adjusted our MR model through a multivariable MR framework accounting for the pathway linking the 25(OH)D SNPs with childhood sunburn episodes and skin colour (Table 4). The effect estimates for melanoma (OR 1.16 [0.94 to 1.43]) and SCC (OR 1.08 [0.89 to 1.31]) in the multivariable model increased very slightly, though the confidence interval overlapped the null. For SCC, the multivariable adjusted estimate for vitamin D showed minor attenuation towards the null (OR 1.15 [0.99 to 1.32]). We note that the strength of the adjustment is hindered by the lack of accuracy in self-reported data and we also omitted the time spent outdoors phenotypes due to lack of power.”

Rev#1 More minor points

Q4. In the abstract the sentence “For basal cell carcinoma, the positive association from standard MR analysis was eventually attenuated after correcting for pigmentation-related variables” appears to contradict the sentence before.

R: We have now fixed this sentence. It now reads:

“Our findings are broadly consistent with previous MR studies indicating no relationship, apart from ovarian cancers (OR 0.89; 95% C.I: 0.82 to 0.96 per 1 SD change in 25(OH)D concentration) and basal cell carcinoma (OR 1.16; 95% C.I.: 1.04 to 1.28). However, after adjustment for pigmentation-related variables in a multivariable MR framework, the BCC findings were attenuated.”

Q5. On line 170; the multivariable MR approaches used here can only evaluate the presence of pleiotropy that acts through sun exposure. This should be made clear for readers unfamiliar with the method.

R: We have now clarified this in text at the end of the Introduction section:

“Multivariable MR approaches were used to account for related exposures, e.g. skin tanning and pigmentation that may **act through** sun exposure on skin cancers.”

Response to Reviewer #2 (Remarks to the Author): Expert in Mendelian Randomization and vitamin D associations, signed by Despoina Manousaki, MD, PhD

The study by Ong et al is an MR revisiting the causal role of vitamin D in 10 cancers, using an increased set of 25(OH)D SNPs as MR instruments, following a GWAS on the recently released UKBB 25(OH)D data. These SNPs explain a larger portion of the variance in 25(OH)D compared to the 6 variants used as instruments in previous MR studies. The results are largely compatible with previous MR studies, showing evidence for a causal role of vitamin D only in ovarian cancer. For some cancers, the confidence intervals of the MR estimates are tighter compared to previous studies. This is the first study reporting MR results on the association of vitamin D with endometrial cancer.

The design of the study has many strengths: the authors used different MR methods, each one with its own assumptions, to control for pleiotropy. They used multivariable MR to test the effect of traditional risk factors such as exposure to sun and skin pigmentation in certain types of skin cancers. Finally, they controlled for sex-specific effects, and reverse effects of cancers on vitamin D levels.

Rev#2 Major points:

Q) My main point of critique is why the authors did you not use the already published GWAS on 25(OH)D in UKBB (Revez et al Nat Comm 2020 or Manousaki et al AJHG 2020). Both GWAS reported a much larger number of independent variants (N~ 140), explaining a much larger portion of the variance in 25(OH)D levels than the 4% reported by the 74 SNPs of the present GWAS. For instance, in the Revez et al GWAS, the SNPs explain up to 13% of the variance in 25(OH)D). Also, these GWAS adjusted for vitamin D supplementation which was not included as a covariate in the GWAS performed for this study. The authors could rerun the MR analyses, and compare their original findings with the results using the SNPs and the respective betas from one of the two GWAS.

R: Thank you for bringing to our attention these recently published and parallel studies on vitamin D. Whilst the aim of our study (causal inference) differed from those in the aforementioned articles (vitamin

D GWASs to discover additional loci impacting 25(OH)D concentration), we concede that the SNP estimates on 25(OH)D obtained from our study might be less accurate for not accounting for vitamin D supplementation in the GWAS analysis. Before we address the issue of the number of SNPs, we considered the impact of accounting for supplementation in the vitamin D GWAS.

Assessing differences due to adjustment on vitamin D supplementation use

We first regressed the estimated 25(OH)D SNP effect size from the Revez et al. and Manousaki et al. onto our 25(OH)D GWAS dataset. We found that the difference in the resultant GWAS effect sizes (with/without adjustment for supplementation) were negligibly small, as shown in the scatter plots in Appendix Figure 1 and Appendix Figure 2 revealing very strong correlation ($r^2 \sim 1$) of the 25(OH)D effect sizes. Whilst our analysis did not account for supplementation use, the strong correlation between these datasets suggests that this limitation is not impacting our results (see Appendix Table 3) - from the identical effect sizes for each SNP, the derived MR results using those variants were very similar.

Evaluation of differences in proportion of 25(OH)D variance explained by SNPs between studies

We re-estimated the variance explained by SNPs on 25(OH)D using the $\sum 2pq\beta^2/\text{Var}(25(\text{OH})\text{D})$ formula (where p refers to the MAF of the SNP instrument, $q=1-p$, β is the SNP effect size on 25(OH)D; summing across all the 25(OH)D SNP instrument and $\text{var}(25\text{OHD})$ is assumed to be one, since the trait is standardized into $N(0,1)$ z-scores) to assess whether the curated sets of independent SNPs from Revez et al. and Manousaki et al. explained more phenotypic variation than the set used in our present analysis. Based on the 134 SNPs reported in Revez et al. and the 138 SNPs from Manousaki et al., the proportion of phenotypic variance (on 25(OH)D concentration) explained by SNPs was estimated to be approximately 0.05 in Revez et al., and 0.08 in the Manousaki et al. dataset (including rare variants).

We respectfully note that “13% of the variance in 25(OH)D” in Revez et al. may be referring to the SNP-based heritability (estimated based on an infinitesimal model assuming small contribution of heritability from each SNP) instead of the proportion of phenotypic variance recovered from only genome-wide significant variants using the above formula (i.e. those likely to fulfill strong instrument criteria, suitable for use in MR analyses).

Furthermore, our analysis used a stricter LD threshold for LD-clumping than used in the LD-clumping or conditional model (GCTA-COJO) of Revez et al and Manousaki et al. We used $r^2=0.001$ based on the recommended threshold for MR

analyses(<https://mrcieu.github.io/TwoSampleMR/articles/exposure.html>); for reference Revez et al used $r^2=0.01$ and Manousaki studies used GCTA-COJO.

To allow for a direct comparison with our MR estimate, we re-clumped the SNP sets extracted from both of these studies using the $r^2=0.001$ LD threshold. When this was done, the estimated variance explained by the genome-wide significant SNPs was similar across the studies (~ 0.04 , see Appendix Table 3 (resulting in 83 independent SNPs in Revez et al. and 59 in Manousaki et al.). We finally provide extra reassurance to the reviewers that our findings remain consistent using SNP sets from Revez et al. as the MR estimates derived from the Revez et al. SNPs were not meaningfully different. Here we have attached a table (see Appendix Table 4) highlighting the comparison of MR findings across both SNP instruments, showing that the results obtained using the 143 variants were very similar to our original findings (79 SNPs). The confidence intervals on estimates were narrower with the larger number of SNPs but this cannot be definitively established as indicating a more certain result because the additional SNPs (143 compared with 79) are extra SNPs which exhibit small but detectable correlations with other SNPs which are already in the model. That is, the extra SNPs are those that have r^2 values such that $0.01 > r^2 > 0.001$ and these may simply represent double counting of what is in fact a single signal at a locus. Hence, in our revised manuscript, we only report the estimates from the clumped set (max LD $r^2=0.001$) of Revez et al. SNPs against ours. The newly added table is now available as Supplementary Table 6 in the revised manuscript.

Changes made to manuscript main text

We have now acknowledged the recently published GWASs in the introduction section of our manuscript: “The GWAS on 25(OH)D in the UK Biobank is approximately four times larger than any previous study and would be expected to dramatically increase the number of candidate 25(OH)D instruments for the multi-SNP approach (see also the very recently published 25(OH)D GWASs (*citation to Revez et al. 2020; Manousaki et al. 2020 added here*)), leading to a well-powered MR analysis while enabling a wider arsenal of MR techniques to address issues of horizontal pleiotropy, assess mediating pathways, and control for genetic heterogeneity among instruments.”

In the Sensitivity analyses part of the Result section, we added:

“We finally evaluated our MR associations using the recently published Revez et al. 25(OH)D genetic instruments that accounts for vitamin D supplementation use - our findings were essentially unchanged (Supplementary Table 6).”

As well as the methods section:

“We also evaluated the impact of adjusting for vitamin D supplementation use by comparing our MR findings with those derived using the Revez et al. SNP instruments.”

Rev#2 Minor points:

Abstract

Line 90: capitalize Mendelian

R: This has now been fixed.

Line 94: unified MR analysis.

R: This has now been fixed.

Line 99: “Unlikely to be a strong risk factor”: I would replace this by “causal risk factor”, since MR is a method for causal inference.

R: We have replaced the term with “causal risk factor” as suggested.

Line 130: “measured to determine vitamin D status”: you could precise “whose serum levels are measured”

R: We have reworded the sentence to make it clearer.

“Vitamin D, whether produced in the skin or consumed, undergoes 2 hydroxylation steps to produce the active form. The first of these produces 25-hydroxyvitamin D (25(OH)D) which can be measured to determine vitamin D status”

Line 136: “examine the issue”: I would say “to investigate the causality of 25(OH)D levels in cancer”

R: We have replaced the term with “investigate whether genetic evidence support a causal role for 25(OH)D levels on cancers”

Line 139: In line with the previous comment, I would say : “Into a potentially causal relationship”

R: We have replaced the term “some insights into the relationship” with “some insights into a potentially causal relationship”.

Line 148-149: I would add: " if more instruments were available, these would explain a larger portion of the variance of 25(OH)D, while recently..."

R: The sentence now reads: "While the biological pathways linking these variants to serum 25(OH)D are generally well understood, if more instruments were available these would explain a larger portion of the phenotypic variance in 25(OH)D, while recently developed multi-instrument based MR sensitivity analyses (such as MR-Egger or median-based approaches) would provide additional assurance that the MR assumptions are not violated"

Line 243: "alternative sensitivity MR models": I would rephrase this as follows: "using various MR methods and sensitivity analyses"

R: We have rephrased the sentence as suggested.

-A statement about power analysis results in the Results section is missing (although this power analysis appears in the supplement).

R: A statement on the power analysis results have been added to the first paragraph of the result section. "Based on the estimated proportion of 25(OH)D variance explained by SNP of 4.0%, the power to detect moderate effect sizes (OR of 1.2 or more per 1 standard deviation (SD) increase in 25(OH)D level) is adequate for most cancers (Supplementary Table 2)."

Line 254, and line 317-318: The authors mention scales: what is the interpretation of a 1SD change in the exposure, and how does that interpret into nmol/l of measured 25(OH)D? Also, how does this compare to changes in serum 25(OH)D as an effect of supplementation? It would be clinically relevant to provide this information. I see a small note in a legend of Table 2 (1 SD corresponds to 20nmol/l change in 25(OH)D, but no mention in the main text.

R: We have added the conversion of 1 SD being equivalent to a 20 nmol/L increase in 25(OH)D at the end of the first paragraph in the result section for better clarity.

"In the UKB, a 1 SD increase in 25(OH)D levels roughly translates to a 20 nmol/L increase in serum 25(OH)D level which is easily achievable with a moderate supplement dose (e.g. ~1000 IU/day) [ref 27]"

Added reference:

27. Institute of Medicine (US) Committee to Review Dietary Reference Intakes for Vitamin D and Calcium. *Dietary Reference Intakes for Calcium and Vitamin D*. (National Academies Press (US), 2011).

Line 269: Please spell out KC and line 294 spell out EC (it's the first mention of these abbreviations).

R: We have added the abbreviation (KC) to where it was first mentioned. And removed the EC abbreviation with the full word “endometrial cancer” since “EC” was only used once throughout the text.

Line 311: Here is the first mention of mode-based method. You need to at least mention all 4 MR methods somewhere in the results before discussing them.

R: We have spelt out the models at the sentence where it was first mentioned while providing a direct reference to the supplementary methods section.

“Evaluation of MR associations under various other MR models (MR-Egger, MR PRESSO, MR mode- and median-based models; see Supplementary methods) showed consistency with the original inverse variance weighted (IVW) estimates, providing evidence against bias due to weak instruments or directional pleiotropy.”

Table 3: Please add the references for the previous MR studies from which the presented estimates were extracted.

R: We apologize for the lack of reference. References for each of the previous MR studies quoted in Table 3 have now been added alongside their corresponding “cancer outcome” column.

Response to Reviewer #3 (Remarks to the Author): Expert in GWAS and MR of vitamin D and cancer

This is a large scale MR study, aiming to test causality of association between 25(OH)D and the risk of several types of cancer. The paper represents a large effort, and conclusions appear largely justified by the data. I have some concerns related to the reporting from the new instrument for 25(OH)D.

Q1) The group has run GWAs on 25(OH)D, presumably in parallel with an other GWAS using this same data from the UKB, which was published some time ago. I appreciate that the published GWAs results (ref 35) may not have been available to the authors at the time of writing this manuscript, however, I was surprised to see running a GWAS in UKB introduced as a hypothetical possibility in the introduction rather than being included with a statement supported by a proper reference (Reference only included in the discussion).

R: At the time when we first conducted the analyses, none of the GWAS studies on vitamin D was published using the UK Biobank data, which we hence were not able to directly use their findings for our MR analysis. We have now amended part of the introduction for the manuscript, to acknowledge these parallel 25(OH)D GWAS studies and expanded on our sensitivity analyses showing that our conclusions remain largely unchanged using SNPs curated from those datasets despite minor differences in our study designs (fitted covariates etc). [See also: our response to Reviewer2]

The relevant sentence in the introduction section now reads:

“The GWAS on 25(OH)D in the UK Biobank is approximately four times larger than any previous study and would be expected to dramatically increase the number of candidate 25(OH)D instruments for the multi-SNP approach (see also the very recently published 25(OH)D GWASs ^{19,20}), leading to a well-powered MR analysis while enabling a wider arsenal of MR techniques ^{21–25} to address issues of horizontal pleiotropy, assess mediating pathways, and control for genetic heterogeneity among instruments.”

Q2) As a key strength of the study they indicate the ability to test for bias by pleiotropy. However, to do this, they first introduce obvious pleiotropy using the new variants. There is a notable lack of transparency with respect to what this new instrument contains (only RS numbers presented, while no gene names are included) but from the formal GWAs I know it includes several well known highly pleiotropic variants such as APOE, GCKR, and CETP, which are unlikely to have a primary role related to 25(OH)D.

R: On annotation of gene names

Given the overall theme of the paper focused on causal inference, we had not previously allocated space in the manuscript to biological interpretation of the newly discovered 25(OH)D variants. To assist readers with interpretation, we have now added the gene names for variants or genes based on closest proximity derived through --annotate in PLINK v1.9b. This information is now available in Supplementary Table 9.

The advantage on pleiotropy assessment using more 25(OH)D variants

Whilst we cannot exclude the possibility of residual pleiotropy driven by a subset of the variants, in practice our results were predominantly null for most cancers - that is, if there really were signals driven by a few pleiotropic SNPs, then these signals were exactly cancelled out by other SNPs (such that the overall result was null).

Furthermore, in circumstances where some of the newly discovered 25(OH)D variants act via pigmentation or related pathways, then our multivariable MR analysis helps account for this. Finally, as discussed below, we have included the results for both a small set of biologically well-supported 25(OH)D SNPs as well as a larger set of SNPs (where the larger set explains more variation in 25(OH)D). As we discuss in the manuscript, one advantage of the larger set is that we are able to employ a range of MR techniques which relax the assumptions necessary for reliable inference from MR (e.g. median, mode, MR Egger). The MR estimates derived based on the 6 SNPs identified from the SUNLIGHT 25(OH)D GWAS were presented alongside the 79 SNP results, available as Table 2 from the main text (also attached as Appendix Table 5) - showing that MR findings from the 79 SNP analysis support the original 6 SNP findings, but with much better precision.

Q3) I do not see the use of the new instrument as a particular advance with this study, as the earlier studies would have been likely to be notably less affected by related biases due to clearly defined roles of the variants in the metabolic vitamin D pathway. Therefore, the title of the paper is misleading in making notes about an ‘improved instrument’ and rather than using this type of opinionated description, my suggestion would be to stick to the facts and e.g. describe the instrument by the number of SNPs.

Given uncertainties with the new instrument, I propose the authors to repeat the analyses using the old instrument. This could be informative, given the sample size for participants with cancer in their study is somewhat larger than in the earlier studies. This would also allow them to meta-analyses across any independent studies presenting information for relevant SNPs , allowing them to maximise the sample size where possible.

R: We agree with the reviewer that for a trait like vitamin D, where there is a reasonable proportion of variance explained from only a handful of vitamin D-associated SNPs (especially where those SNPs have well understood biological roles in the synthesis and metabolism of vitamin D), then an MR analysis focusing on just those SNPs can be informative. We apologize for not making this clear in the manuscript previously but we had previously incorporated and reported both the MR estimates based on the 6 well-known vitamin D instruments, and the more polygenic set of 74 SNPs discovered in the UKB vitamin D GWAS. We have now added a sentence at the start of the result section to make this clearer to readers.

“The estimated OR per 1 SD increase in genetically predicted 25(OH)D on cancer risks using the traditional 25(OH)D SNP instruments (n=6) and the larger set of SNP instruments identified from the UK Biobank (n=79) are shown separately in Table 2.”

Whilst we also agree that the vitamin D findings derived from the old instruments (6 SNPs) have the advantage of building on SNPs with well understood effects on vitamin D biology, a common criticism for MR studies with few instruments is their vulnerability to false positive findings. Because power for many of the pleiotropy assessments in MR to detect bias increases when more variants are included, we felt that findings from the old instruments will always be subjected to strong criticism on inadequate treatment of horizontal pleiotropy and the lack of power to detect them. This is in part also the main motivation for this study, to “revisit” these vitamin D associations

with cancer, and show that having more vitamin D SNPs would enable better assessment of genetic pleiotropy in a standardised study design across multiple cancers, and provide more robust claims on its (null) association with most cancers.

We understand the reviewer's concern on the suggestive wording ("improved instrument") in our manuscript title; hence we have removed this term from our manuscript title. The Manuscript title now reads: **"Revisiting the association between vitamin D and cancer susceptibility: a unified Mendelian randomization analysis"**

Q4) In power calculation tables (supplementary 2), it is not made clear what is the estimated difference in 25(OH)D that relates to the proposed ORs. Please also clarify units elsewhere, and ensure it is clear to the reader what kind of difference in 25(OH)D (optimally in units such as nmol/l) relates to each of ORs across all tables and figures.

The power calculations for two-sample MR estimated in Supplementary Table 2 were derived based on the power to detect an association at the proposed OR per one SD increase in 25(OH)D concentration. The 25(OH)D measurements were standardised via an inverse-gaussian transformation prior to the GWAS analysis; however one SD increase is equivalent to a 20 nmol/L increase in 25(OH)D based on the UK Biobank biomarker data. We have now added the relevant measurement in the table caption for each corresponding table reporting these MR-derived ORs. We have added a similar description to the end of the first paragraph in the result section to aid the readers:

"Based on the estimated proportion of 25(OH)D variance explained by SNPs of 4.0%, the power to detect moderate effect sizes (OR of 1.2 or more per 1 standard deviation (SD) increase in 25(OH)D level) is adequate for most cancers (Supplementary Table 2). In the UKB, a 1 SD increase in 25(OH)D levels roughly translates to a 20 nmol/L increase in serum 25(OH)D level which is easily achievable with a moderate supplement dose (e.g. ~1000 IU/day)."

- End of point-by-point response -

--END OF LETTER--

REVIEWERS' COMMENTS

Reviewer #1 (Remarks to the Author):

I am happy that the authors have addressed all of my previous comments. It appears however that the description of the multivariable MR at the beginning of the methods section (line 429) has not been fully updated to reflect the new approach and so should be updated.

Reviewer #2 (Remarks to the Author):

The authors have addressed adequately the comments of my previous review. The only comment I have is on the title, specifically the term "unified" Mendelian randomization. I think it could cause some confusion in regards to the MR method. Could the authors think of any alternative terms?

Reviewer #3 (Remarks to the Author):

no further comments

RESPONSE TO REVIEWERS' COMMENTS

Reviewer #1 (Remarks to the Author):

I am happy that the authors have addressed all of my previous comments. It appears however that the description of the multivariable MR at the beginning of the methods section (line 429) has not been fully updated to reflect the new approach and so should be updated.

Response: We apologize for the mistake. The text in the method section have been corrected to reflect the new approach (i.e using only 3 traits in the MVMR analysis reported in the main text) applied to the manuscript:

In methods, under [Obtaining instruments for skin exposure traits], we now have

“For the multivariable MR analysis on skin-related cancers (details of the analysis expanded below), we also estimated the joint effect of 25(OH)D instruments along with episodes of childhood sunburn and skin colour on risk of skin cancers. These phenotypes, obtained through questionnaires, were available in the UK Biobank. In brief, we obtained self-reported data on: episodes of childhood sunburn from 331 020 and skin colour from 433 288 WB UKB participants with suitable genetic data. The complete list of the sun exposure traits considered for the multivariable MR analysis together with the phenotype definitions and preparation for GWAS on these traits are described in greater detail in Supplementary methods.”

Reviewer #2 (Remarks to the Author):

The authors have addressed adequately the comments of my previous review. The only comment I have is on the title, specifically the term "unified" Mendelian randomization. I think it could cause some confusion in regards to the MR method. Could the authors think of any alternative terms?

Response: We agree that the term “unified” might imply a novel MR method, which can be confusing to the readers. We have removed such wording, and reworded our manuscript title from

“Revisiting the association between vitamin D and cancer susceptibility: a unified Mendelian randomization analysis”

to

“A comprehensive re-assessment of the association between vitamin D and cancer susceptibility using Mendelian randomization”

Reviewer #3 (Remarks to the Author):

no further comments

Response: We are grateful that the reviewer for finding our revisions satisfactory.